# Investigation of Transmission and Evolution of PEDV Variants and Co-Infections in Northeast China from 2011 to 2022

**DOI:** 10.3390/ani14152168

**Published:** 2024-07-25

**Authors:** Feipeng Zhao, Xin’ao Ma, Jianfeng Yang, Zhiying Wei, Jiaxuan Li, Yanping Jiang, Wen Cui, Zhifu Shan, Lijie Tang

**Affiliations:** 1College of Veterinary Medicine, Northeast Agricultural University, Harbin 150030, China; 13244538010@163.com (F.Z.); mxa13188921810@163.com (X.M.); 15176703037@163.com (J.Y.); wzy1374976073@163.com (Z.W.); lijiaxuan.1993@163.com (J.L.); jiangyanping8198@163.com (Y.J.); cuiwen@neau.edu.cn (W.C.); 2Northeast Science Observation Station for Animal Pathogen Biology, Ministry of Agriculture and Rural Affairs, Harbin 150030, China

**Keywords:** porcine epidemic diarrhea virus, co-infection, S gene, epidemiology

## Abstract

**Simple Summary:**

Simple Summary: Porcine epidemic diarrhea virus (PEDV) is a rapidly evolving pathogen responsible for outbreaks in pig herds globally. Mutations in the S protein of PEDV have resulted in new variants and a reduction in vaccine efficacy. This epidemiological survey in northeast China from 2015 to 2022 examined the prevalence of enteroviruses co-infected with PEDV and the structural mutations in the spike (S) protein resulting from mutations of the S gene from 2011 to 2022. Notably, two distinct mutations in the emerging 2022 HEB strain were identified. These findings enhance the understanding of PEDV co-infection and genetic evolution in northeast China.

**Abstract:**

Porcine epidemic diarrhea virus (PEDV) is a rapidly evolving virus that causes outbreaks in pig herds worldwide. Mutations in the S protein of PEDV have led to the emergence of new viral variants, which can reduce vaccine immunity against prevalent strains. To understand the infection and variation pattern of PEDV in China, an extensive epidemiological survey was conducted in northeast China from 2015 to 2022. The genetic diversity of enteroviruses co-infected with PEDV and the PEDV S gene was analyzed, common mutation patterns that may have led to changes in PEDV virulence and infectivity in recent years were identified, and structural changes in the surface of the S protein resulting from mutations in the PEDV S gene from 2011 to 2022 were reviewed. Of note, two distinct mutations in the emerging 2022 HEB strain were identified. These findings provide a basis for a better understanding of PEDV co-infection and genetic evolution in northeast China.

## 1. Introduction

Porcine epidemic diarrhea (PED) is a highly fatal infectious disease of piglets caused by the porcine epidemic diarrhea virus (PEDV) [1,2]. It has similar clinical signs to transmissible gastroenteritis (TGE): both present as acute diarrhea; and was first identified in the United Kingdom in 1971 and has since spread throughout the world. The first case of PEDV-induced diarrhea in pigs in China was reported in 1973, and no major outbreaks have occurred since PEDV spread. The first outbreak of acute swine diarrhea in Japan was confirmed to be caused by PED in 1983, and between 1993 and 1994, an outbreak in Japan resulted in the death of 14,000 pigs [3]. Mortality rates in suckling piglets ranged from 30% to 100% [4]. Since then, the virus has spread throughout Asia and can be controlled with a vaccine [5]. But after 2006, new strains of PEDV were spreading and circulating in pigs that had been immunized with the original PEDV vaccine [6], suggesting that the original vaccine had lost its immunity to the new strain and that new strains had begun to circulate. At the end of 2010, a PEDV outbreak occurred in several pig-producing provinces in southern China [7]. Subsequently, the disease spread to several provinces in China, causing significant financial losses to the pork industry.

PEDV belongs to the *Alphacoronavirus* genus in the subfamily of *Coronaviridae*. It is a non-segmented, capsid, single-stranded RNA virus with a positive-stranded genome of approximately 28,000 nucleotides [6]. The genome contains seven open reading frames and non-coding regions situated at the 5′ and 3′ ends. Two-thirds of the 5′ end of the PEDV genome encodes the polyproteins ORF1a and ORF1b, which play important roles in viral replication and suppression of host innate immunity [8,9,10,11]. The sequence’s 3′ end of the sequence encodes several proteins, including the spike protein (S), envelope protein (E), membrane protein (M), nucleocapsid protein (N), and an accessory protein to ORF3 [12].

The S protein of PEDV has a critical function in viral receptor binding, membrane fusion, and host cell invasion. It also carries the major antigenic epitope that induces the production of neutralizing antibodies in the host. The S protein consists of two sections: the S1 region at the N-terminus; and the S2 region at the C-terminus [13]. Recent studies have shown that the S1 region of the S protein plays a key role in determining the pathogenicity of PEDV. This region is also responsible for facilitating the attachment of viral particles to cell surface receptors [14]. The S1 region is characterized by considerable variability and represents the principal functional element of the S protein. Upstream of the S1 region is the signal peptide (SP) and downstream of the SP are two S1 subunits, S1 N-terminal (NTD) and S1 C-terminal (CTD). The region is relatively conserved, with the transmembrane region (TM) and cytoplasmic tail (Cyto) positioned at the C-terminus. Recent studies have suggested that the S1-NTD may be a potential region associated with PEDV virulence [15]. Previous studies have shown that the PEDV S1 region is essential for virus–host recognition [16]. The S1 region of all coronaviruses contains the receptor binding domain (RBD), which recognizes cellular receptors to facilitate viral entry [17]. The classic, weak PEDV strain CV777 and the mutant strain GHGD-01 do not differ significantly in the ability to bind porcine aminopeptidase N (pAPN), the primary receptor of PEDV. However, GHGD-01 is more capable of recognizing glycosylated proteins [16]. This may explain the difference in virulence between the two strains.

Mutations in the S protein of PEDV have resulted in an increase in viral genetic diversity, which has impacted the global PEDV epidemic [18]. Phylogenetic analysis of S sequences classified PEDV into two genotypes, GI and GII. It is notable that the classic strain CV777, which was prevalent in the early years, belongs to the GI subgroup, whereas globally prevalent of PEDV strains after 2010 are dominated by strains in the GII subgroup [19]. Moreover, the majority of PEDVs identified in China at this stage are of the GII subtype [20]. In 2013, attenuated strains of PEDV, namely S-INDEL and OH851, were initially identified in the USA, and have subsequently been reported in other countries [21]. Wang documented the appearance of three attenuated PEDV strains, namely non-S-INDEL, ‘CV777’ S-INDEL, and ‘US’ S-INDEL, within a concurrent transmission in Chinese pigs [7,11,22]. In 2018, Su discovered that deleting S1 NTD in PEDV caused enhancement when co-infected with PEDV in piglets [23]. However, natural S1 NTD deletion strains in the wild frequently co-infects with intact PEDV [23,24,25,26,27]. Recent epidemiological evidence indicates that the S gene of PEDV is undergoing rapid mutations. Mutations in the S protein of the current strain in China have resulted in increased viral activity towards sialic acid binding, compared to the classic PEDV strain [28]. Therefore, understanding the pattern of insertion and deletion mutations in the S protein is crucial data for better monitoring of PEDV epidemics and re-emergences in Asia, Europe and North America.

In addition to PEDV, several other porcine enteric viruses are responsible for swine diarrhea, including *transmissible gastroenteritis virus* (TGEV), *Porcine Teschovirus* (PTV), *Porcine Sapelovirus* (PSV), *Porcine Enterovirus* (PEV), *Porcine Rotavirus* (PRoV), *Porcine Astrovirus* (PAstV), *Porcine Circovirus* (PCV), *Porcine Bocavirus* (PBoV), *Porcine Kobuvirus* (PKV), *Torque Teno Sus virus II* (TTSuV-2), and *Porcine Delta coronavirus* (PDCoV). Recent data suggest that PEDV is the primary cause of viral diarrheal disease in pigs throughout China [29]. Viral diarrheal diseases caused by porcine enteric pathogens, as well as co-infections with multiple enteric pathogens are prevalent in piglets with diarrhea [30,31,32]. It has been reported that PEDV and PBoV are co-infected more frequently [33]. Chen found that 73% of the tested samples had co-infection with two to nine pathogens [34]. Testing of porcine diarrheal samples from Sichuan showed that 12.5% of samples tested presented co-infection with PKV, PAstV, and PToV [35]. Chang et al. reported severe multiple infections with PEDV and TGEV [36]. Viral variation and simultaneous co-infection with multiple pathogens complicate the control of diarrheal disease in pigs infected with PEDV. However, there is a lack of information on the co-infection of piglets with diarrheal diseases. Further research on co-infecting PEDV with other pathogens is required.

The Northeast region of China is also a major swine breeding area, but it differs greatly in terms of climate, temperature and precipitation compared to the Southern part of the country. There are significantly long, cold seasons in winter, which may have an impact on the PEDV emergence, transmission and variation. Therefore, this study investigated the emergence, transmission, and evolution of PEDV variants as well as their co-infection with other enteric pathogens in Northeast China from 2011 to 2022. A total of 327 samples of piglet diarrhea collected from northeast China were analyzed for co-infection, S gene mutation pattern analysis, phylogenetic analysis and protein homology modelling in order to determine the prevalence and changing patterns of PEDV in Northeast China.

## 2. Materials and Methods

### 2.1. Sample Collection and Gene Acquisition

This study obtained 327 fecal samples from piglets under 10 days old suffering from diarrhea between January 2015 and March 2022, from four northeast Chinese provinces: Heilongjiang (*n* = 90), Jilin (*n* = 81), Liaoning (*n* = 85) and the Inner Mongolia Autonomous Region (*n* = 71). All samples were stored at −80 °C.

### 2.2. PCR Detection of PEDV and Other Pathogens

RNA extraction and cDNA synthesis followed the protocols described by Wang et al. [37]. DNA extraction was conducted based on the protocol by Qi et al. [38]. To detect PEDV and 11 other enteroviruses, we used a nested PCR method based on the design of this study or previous studies [25,29,30,31,37,39,40,41,42] (Appendix A). The enteroviruses tested were PTV, TGEV, PSV, PEV, PRoV, PAstV, PCV, PBoV, PKV, TTSuV-2, and PDCoV.

### 2.3. PEDV S Gene Sequencing and Analysis

To amplify the S gene of PEDV, we used the primers detailed in Appendix A. Each amplified S gene was subjected to Sanger sequencing. The reference genome sequence of the endemic PEDV strain was obtained from NCBI Genbank (revision 247.0) (Appendix A). Sequence analysis was carried out using the EditSeq tool in Lasergene DNASTAR™ 5.06 software (DNASTAR Inc., Madison, WI, USA). Multiple sequence alignments were carried out using the multiple sequence alignment tool of DNAMAN 6.0 software (Lynnon BioSoft, San Ramon, CA, USA). The naming system was used to describe sequence variation [43]. Point mutations detected in PEDV S proteins were counted using GraphPad Prism 8.0 (GraphPad Software, Inc., La Jolla, CA, USA).

### 2.4. Recombination Analysis

Recombination analysis was conducted with RPD5 5.05 software (open-source tools from CBIO, University of Cape Town) to investigate potential recombination events between the PEDV S gene sequences obtained from clinical specimens in this study and those accessible on NCBI. Sample sequences with a *p*-value of up to 0.05 were considered indicative of recombination events across all seven methods implemented in RPD5 according to Martin’s method. Recombinant sequences were removed, and the analysis was repeated until no further recombination events were detected [44]. The map of China was created with Tableau 2023.1 software (Tableau Software, Inc., Seattle, WA, USA).

### 2.5. Phylogenetic Analysis

To conduct a phylogenetic analysis, we retrieved the S gene of PEDV strains from GenBank (revision 247.0) (Appendix A). These nucleotide sequences were then used to generate a neighbor-joining phylogenetic tree for the S gene using the ClustalX comparison tool in the MEGA 6.06 software [45]. The neighbor-joining phylogenetic tree was constructed using a p-distance model and 1000 bootstrap replicates [46]. The phylogenetic trees were annotated using iTOL 5.2 software (European Molecular Biology Laboratory, Heidelberg, Germany), an online tool for modifying and landscaping phylogenetic trees [47].

### 2.6. Molecular Modeling and Analysis of the PEDV S Protein

The study identified the major amino acid sequences of PEDV S from 2011 to 2022 that were consistently found (named 2011_pedv-strain, 2012_pedv-strain, 2013_pedv-strain, 2014_pedv-strain, 2015_pedv-strain, 2016_pedv-strain, 2017_pedv-strain, 2018_pedv-strain, 2019_pedv-strain, 2020_pedv-strain, 2021_pedv-strain and 2022_pedv-strain) (Appendix A). The conformation of the mutant PEDV S protein was ascertained. The tertiary structure of the S region was predicted through the use of Swiss-Model (https://swissmodel.expasy.org/, 16 January 2023), an open-source modelling server from the Swiss Institute of Bioinformatics [48]. The stinger protein of the human coronavirus NL63 (PDB ID: 5SZS) was utilized as a template for the formation of the S-monomer tertiary structure.

### 2.7. Statistical Analysis of Correlation

The correlation between PEDV infection and other pathogens, including TGEV, PSV, PEV, PRoV, PAstV, PTV, PBoV, PKV, PDCoV, TTSuV-2, and PCV, was analyzed using the chi-square (χ^2^) test in IBM SPSS Statistics version 22.0 (IBM^®^ SPSS Inc., Chicago, IL, USA). The significance level for all analyses was 5% with a 95% confidence interval. Differences in values were considered statistically significant or highly significant if the associated *p*-value was <0.05 or <0.01, respectively.

## 3. Results

### 3.1. Co-Infection of PEDV with Multiple Pathogens in Piglets with Diarrhea

Of the 327 diarrhea disease samples collected from piglets, the following pathogens were identified by nested PCR: PEDV at 70.9% (232/327), PRoV at 33.33% (109/327), PTV at 30.28% (99/327), PCV at 29.66% (97/327), PAstV at 12.84% (42/327), PKV at 6.72% (22/327), TGEV at 6.42% (21/327), PSV at 3.98% (13/327), PEV at 3.36% (11/327), PBoV at 0.92% (3/327), PDCoV at 1.22% (4/327) and TTSuV-2 at 0% (0/327) (Appendix A; Figure 1a). Figure 1b shows that 66.97% (219/327) of the samples contained at least two pathogens. In addition, 29.05% (95/327) of the samples showed the presence of 3–5 different pathogens.

Of the 232 PEDV-positive samples, PRoV, PTV and PCV were the most frequently co-infected viruses, with rates of 40.09% (93/232), 34.91% (81/232) and 32.32% (75/232) (Appendix A; Figure 2). Within all tested enterovirus samples, the rate of mixed infections with PEDV was notably high, ranging from 95.45% to 33.33%. Additionally, the tested samples exhibited a considerable percentage of mixed infections with PKV and PRoV, in addition to PED (Figure 2). Chi-square analysis showed that none of the viruses had a significant correlation with PEDV infection (Appendix A).

### 3.2. PEDV S Gene Sequencing and Analysis

The research successfully collected 155 PEDV S genes from PEDV-positive samples taken from piglets in northeast China between 2015 and 2022. The nucleotide and amino acid homology of the 155 S genes varied from 96.3% to 99.9% and 98.2% to 92.7%, correspondingly. The nucleotide and amino acid homology of all sequences with strain CV777 was 90.6%~91.3% and 90.1%~92.3%, respectively. To further explore the evolution of the S protein of the PEDV strain, the deduced amino acids of 732 S genes in NCBI (2011–2022) were compared with those in the study (28 in 2015, 21 in 2016, 23 in 2017, 15 in 2018, 20 in 2019, 10 in 2020, 14 in 2021 and 24 in 2022), using the S protein of CV777 as a reference. The study comparatively analyzed the mutation patterns of 887 S genes. Among the 887 S protein mutations (using the total number of all sequences), the most common mutation pattern was S58-S58 QGNV, S156-S156 H/Y, S359-S359 NMRS insertion, and S328 S358 S366 S772 S776 S782 S814 S904 S967 S973 S981 S1032 S1052 S1175 S1201 S1241 S1268 S1306 substitution and deletion of S1202. Additionally, two distinct mutation patterns were identified in the 2022-HEB variant: the replacement of S1274 P with S and S974 S with A. Amino acid mutations were observed in the sialic acid binding active site S1 protein NTD (amino acids 9-433) and the receptor binding domain of S1 protein (RBD, amino acids 501-629) of prevalent PEDV sequences from 2011 to 2022. The data collected highlight that the PEDV S gene is frequently mutated at the S58–S58 position for the QGNV, S156–S156 H/Y and S359–S359 NMRS insertions. Additionally, mutations in the S328, S358, S366, S772, S776, S782, S814, S904, S967, S973, S981, S1032, S1052, S1175 and S1201 positions, as well as substitutions in S1241, S1268 and S1306, and the deletion of S1202, are established mutation patterns among prevalent PEDV strains from 2011 to 2022 (Figure 3a) and the comprehensive overview of the detailed mutations is listed in Appendix A.

The amino acids of 732 prevalent PEDV S genes from 2011 to 2022 in NCBI Genbank were compared to the 155 S genes identified in this study to investigate the evolutionary trends of S proteins in PEDV strains. The results indicated that the rate of PEDV S amino acid mutations has steadily increased since 2011, with mutations primarily concentrated in the S1 region. The rate of amino acid mutations in the PEDV S protein has been rising steadily since 2011, with percentages increasing from 0.71% in 2012, to 3% in 2022. The average amount of amino acid mutations in the PEDV strains rose from 10 in 2012 to 42 in 2022, with new mutation sites being added each year (13 in 2013, 14 in 2014, 15 in 2015, 24 in 2016, 28 in 2017, 29 in 2018, 38 in 2019, 39 in 2020, 40 in 2021, and 42 in 2022) according to Figure 3b. At the same time, the numbers of new variant sites of the PEDV S protein increased year by year (Table 1). This indicates ongoing viral evolution, which can alter the function and structure of viral proteins, affecting the infection rate and pathology of the virus. The insights gained from these mutation sites can assist in determining the genetic variation and evolutionary path of PEDV over time.

### 3.3. Recombination Analysis

A recombination analysis of the PEDV sequences included in the study was performed on the basis that coronaviruses are highly susceptible to recombination, which may reveal the origin and distribution of PEDV and its evolutionary pathways in different geographical locations and time periods. A total of 168 recombination events were observed in all the strains analyzed, with 18 recombination events occurring in non-Chinese endemic strains, with high frequency in Korean, Vietnamese, Spanish and Polish endemic strains, and 75 recombination events occurring in Chinese endemic strains, with a high probability of recombination events pointing to Guangdong and Sichuan. It was subsequently found that strains prevalent in Guangdong were closely related to strains prevalent in Korea and Mexico, and that strains prevalent in Sichuan were more closely related to strains prevalent in Japan, Vietnam and Mexico (Figure 4, Appendix A). Notably, most of the recombination sites of the strains prevalent in non-Chinese regions were concentrated at the head of the S gene, whereas the recombination sites of the strains prevalent in Chinese regions were more widespread, with recombination events occurring in all parts of the S gene (Figure 5, Appendix A). These data suggest that PEDV recombination is more common, that the strains prevalent in China recombine more rapidly than in other countries and regions, and that the recombinant strains are closely related to those prevalent in Korea, Japan and Vietnam.

### 3.4. Phylogenetic Analysis

A phylogenetic tree was constructed to analyze the S genes of 155 PEDV strains identified in this study and 732 reference strains from GenBank for the purpose of phylogenetic analysis. The PEDV strains identified in this study were closely related to the reference strains prevalent in China. Of the 155 total strains, 6 (3.87%) belonged to the GIa subgroup, 0 (0%) to the GIb subgroup, 1 (0.65%) to the GIIa subgroup, 139 (89.68%) to the GIIb subgroup, and 9 (5.80%) were recombinant strains. The phylogenetic tree analysis results indicated an increase in the proportion of subgroup GIIb strains among the prevalent PEDV strains in China over time, while the proportion of GIIa subgroup strains decreased each year from 2011 to 2022, as shown in Figure 6. Furthermore, these analyses revealed the genetic diversity and complexity of PEDV, indicating variation and recombination of PEDV in different regions and time periods. This data may help to better understand the genetic evolution and transmission patterns of PEDV, thereby increasing the effectiveness of PEDV prevention and control strategies.

### 3.5. PEDV S-Protein Modeling Analysis

The analysis of the S protein modelling revealed that the structure of the PEDV S protein underwent changes on an annual basis, exhibiting a discernible pattern of alteration (Figure 7, Table 2). The range of structural variations observed in the PEDV S protein exhibited a gradual fixation, with the neighborhoods of residues 55–64, 113–117, 133, 157–159, 198, and 210, as well as residues 524–527 demonstrating persistent variability. The 310 helix in the corresponding region of the S protein undergoes alterations, including the conversion to a β-turn, displacement of extended strand structures and the emergence and disappearance of 310 helix structures. It is noteworthy that the structure in the vicinity of residues 157–159 undergoes alterations from year to year. This may be the region of the viral S protein that is most important for its functional changes.

## 4. Discussion

PEDV is a notable enteropathogen associated with outbreaks of piglet diarrhea in China, and this study found a high infection rate of 70.9% (232/327) in affected piglets. Despite extensive research into PEDV infection mechanisms and potential vaccine candidates, controlling the disease proves challenging due to co-infection with other pathogens [43]. To gain a better understanding of the frequency of PEDV co-infection with other pathogens in Northeast China, an extensive epidemiological investigation was conducted in this study. We found a high proportion of mixed infections in piglet diarrhea samples from northeast China. Out of the 327 diarrhea samples analyzed, 219 (66.97%) were simultaneously infected with two distinct viruses, and 95 (29.05%) were simultaneously infected with three or more different pathogens. The study revealed that only 2.75% (9/327) of individuals diagnosed with porcine diarrhea disease were infected with PEDV alone. It is noteworthy that no viral infection was detected in 29.05% of the samples we tested. Due to the multitude of clinical causes that can lead to swine diarrhea, we hypothesize that the samples with no detectable viral infections may have been the result of bacterial infections or the result of stress during the feeding process.

The majority of samples showed co-infection with other pathogens in northern China. Sungsuwan et al. found that PDCoV and TGEV competed with each other when co-infected with PEDV. However, an excess of TGEV resulted in an increased replication of PEDV in the host, providing evidence of functional compatibility and evolutionary relationships between CoV viral proteins during co-evolution. These results confirm the interactions of inter-viral co-infection [49], supporting the infection data for PEDV and PDCoV in our study. The prevalence of PRoV positivity in the 327 diarrhea samples collected between 2015 and 2022 in Northeast China varied from 74.50% to 13.51%. This finding is consistent with Chang’s research, which reported a co-infection prevalence of 16.67% for PEDV and PRoV. PRoV has been found to have a higher co-infection rate during outbreaks and epidemics of diseases caused by PEDV. Swine infections of encephalomyelitis commonly involve PTV, which was first identified in China in 2002, and some studies have shown that the virus often occurs in co-infection with other diarrhea viruses [29,50,51].

Guo et al. showed that mixed infections of PCV and PEDV were prevalent in Hebei, China [52]. PCV infection in pigs primarily damages the organism’s immune system [53]. In this study, high infection rates of both PTV and PCV were observed in diarrheic piglets. A greater number of samples were found to be co-infected with PEDV, PTV, and PCV than those which tested negative for PEDV. These findings suggest that co-infection of PEDV with PRoV and PTV may exacerbate clinical signs of diarrhea in piglets. It is worth noting that in this part of the study, PKV-infected samples showed a high rate of mixed infection with PEDV (95.45%), consistent with the data from Su et al. [49]. There may be a potential link between PKV and PEDV co-infection. However, our chi-square analysis of the infection data of the two viruses revealed no significant correlation. For other viruses, we also did not observe a significant correlation with PEDV. This may be influenced by the limited number of samples collected and the sampling environment. Further studies are needed to verify whether PKV infection affects the prevalence of PEDV. The high proportion of viral mixed infections and the variety of mixed viruses in piglet diarrhea samples from northeast China during 2015–2022 suggest that hosts were exposed to multiple viruses, which may have facilitated genetic recombination and evolution among viruses [54]. This highlights the need for purification and an improvement in piglet rearing environments and practices.

To comprehend the changes in PEDV genes, homology modelling of the S protein was conducted, which revealed mutations at specific residues in prevalent PEDV strains between 2011 and 2022. These mutations included residues 55–64, 113–117, and S gene residues 133, 157–159, 198, and 210, resulting in changes in the structure of the NTD region from year to year. This region is associated with silicate-binding activity, which is critical for PEDV’s ability to infect the host [16,20]. Changes at residues 55–64 resulted in a change in the 310 helix in this region to a β-turn, and changes at residues 113–117 and 133 resulted in a displacement of the extended strand structure that would have been present here. In contrast, modifications in the vicinity of residues 157–159 caused irregular alterations in the protein structure. Alterations at residues 198 and 210 resulted in a shift in the position of the extended strand and the appearance or disappearance of the 310 helix. It is possible that this may increase the cellular entry capacity of pandemic PEDV strains in China by altering silicate-binding activity. Annual structural changes occur in the RBD region at residues 524 to 527, which affect the virus’s ability to bind to cells. This section describes a potential mode of mutation for emerging strains. The change from residues 524 to 527 results in a shift in the position of the extended strand in this region. The structure was altered in the strains that were prevalent in 2017 and 2018. In these strains, the extended strand was replaced with the β-bridge and bend region, respectively. This site is also where the virus is neutralized, which affects its ability to be neutralized. The structural mutations of residues 559–567 and 660–661 in the 2011–2013 epidemic strains gradually stabilized over time. These changes in residues 559–567 and 660–661 in the 2011–2013 epidemic strains resulted in shifts in the structure of the extended strand and changes in its number. This suggests a gradual stabilization of the extended strand structure of residues 559–567 and 660–661 of the S protein in the PEDV strains prevalent after 2013. This region may be a signature structure of the strains that became popular after this year. These results corroborate the findings of Chen et al. [55], who observed a steady increase in the number of new mutation sites in the dominant PEDV strains between 2011 and 2022, accompanied by a stabilization of the structural mutations in the viral NTD and RBD. These structures may be potential key sites for the virus. Zhang et al. observed that mutations in the PEDV S protein resulted in an increase in the α-helical structure at residues 55–64 and the replacement of the β-turn by an α-helix at residues 157–164, which altered the virulence, pathogenicity, and immunogenicity of the virus and led to the evolution of the virus from subgroup GI to subgroup GII. Residues 55–64 are specific for recognizing GI strains, while residues 157–164 exhibited specificity in terms of immunogenicity [56]. Our study revealed that the region of PEDV S protein expanded to residues 55–64, 113–117, 133, 157–159, 198 and 210 between 2011 and 2022. Furthermore, fluctuations in these residues on an annual basis may impede the ability to discern the evolutionary characteristics of PEDV. The constant changes in residues 55–64 may indicate that the newly prevalent strains in recent years are diverging from the immunological profile of the classic strain (GI).

The data indicate that the protein structure of PEDV changes at the specified positions on a yearly basis. PEDV is evolving to dispel researchers’ perceptions of the immunogenicity of the classic strains. Our study indicates that the amino acid mutations at 360–363 in the 2022-HEB strain may enhance viral adaptation to the environment. Additionally, structural changes at 553–565 may lead to alterations in viral neutralization activity, which could have significant implications for the development of effective therapies and vaccines [38,56]. Finally, mutations in the S1 region may be responsible for the failure of immunization against recent epidemic strains. Therefore, studying the mutation patterns of the S1 protein can be a crucial starting point in comprehending the worldwide resurgence of PEDV. Our findings highlight the significance of comprehending the mutation patterns and structural domains of the PEDV S protein to establish effective methods for preventing and treating PEDV infection.

To delve into the predominant genotypes and the spectrum of genotypic diversity among the epidemic strains from 2011 to 2022, we developed a phylogenetic tree based on the S proteins of PEDV. A survey conducted by Hong et al. across 17 provinces in China revealed that 92.3% of the PEDV strains circulating during 2020–2021 belonged to the GII subtype, while only 7.7% of the S-INDEL-like strains belonged to the GI subtype [57]. The findings of Shen et al. indicated that the predominant strains of the PEDV GII subtype were present in Shandong Province during the period between 2019 and 2021 [58]. A study by Chen et al. identified a notable mutation in the PEDV S protein in the Xinjiang region of China between 2020 and 2022. This is consistent with our observations. The strains in this region were predominantly GIIa, which differs from our findings. INDEL strains, which have emerged in several countries recently, were not identified in our study. These discrepancies may be attributed to differences in their geographical distribution [59,60].

It is known that sera from PEDV GI and GII subtype strains do not cross-recognize [61]. According to recent research on serotype-specific clinical serum samples, substitutions at positions 55–64 of amino acids are distinct in identifying host GI strains, whilst mutations at positions 157–164 impact the host’s immune specificity to GII strains [56]. Therefore, conducting comprehensive research on the mutations at these sites may pave the way forward for developing a genetic vaccine that effectively prevents and controls both GI and GII genotypes. Recent studies have indicated that conventional PEDV vaccines provide superior protection against GI subtype strains, whereas there are no available vaccines for emerging GII strains that offer effective protection [62,63]. Moreover, the research by Zhang et al. noted that the PEDV GII genotype strain is presently the most commonly encountered strain in China and Korea. The research highlights that globally, GII strains are undergoing more rapid evolution compared to GI strains, aligning with the SARS-CoV-2’s abrupt mutation which enables immune evasion within the host [64]. Our collected data further demonstrate that the percentage of PEDV GII strains is progressively increasing year-on-year, reflecting the heightened urgency for effective measures against GII strain epidemics. Furthermore, the study indicates that certain prevalent strains of PEDV experienced multiple recombination events with vaccine strains in recent years. This suggests that the high prevalence and evolutionary changes in GII strains could be linked to the use of vaccines against the PEDV GII subgroup. This could be another possible explanation for the decreased immune effectiveness of current vaccines against PEDV [56].

Possible links between pig trade and PEDV transmission have been reported by He et al. [65]. Our research established a clear recombination bond between native endemic strains and those originating in Poland, Japan, and Vietnam. Zhang et al. identified Poland, Romania, Japan, Thailand, Mexico and Colombia as countries exporting PEDV in Europe, Additionally, Japan plays a significant role as a major import/export center for PEDV. Moreover, the characteristics of China’s PEDV import/export centers indicate the existence of significant transmission links to 11 provinces [38]. Our PEDV recombination research supports these findings and indicates that the import and export trade of pigs may be a significant contributing factor to the global spread and recombination of PEDV.

## 5. Conclusions

In conclusion, our study provides evidence that PEDV co-infects with 11 enteroviruses. The PEDV strains identified in our study belonged to Chinese endemic strains and showed genetic diversity, with the majority belonging to the GIIb subgroup. Amino acid mutations in the S gene were identified, and a distinct mutation pattern within the S gene was observed, altering the protein structure relative to PEDV CV777. The PEDV strains prevalent in China recombine faster than in other countries and regions, and the recombinant strains are closely related to those prevalent in Korea, Japan and Vietnam. These findings have significant implications for understanding the co-infection, transmission, and genetic evolution of PEDV in the past 10 years, as well as developing novel strategies to control the disease.

## Figures and Tables

**Figure 1 animals-14-02168-f001:**
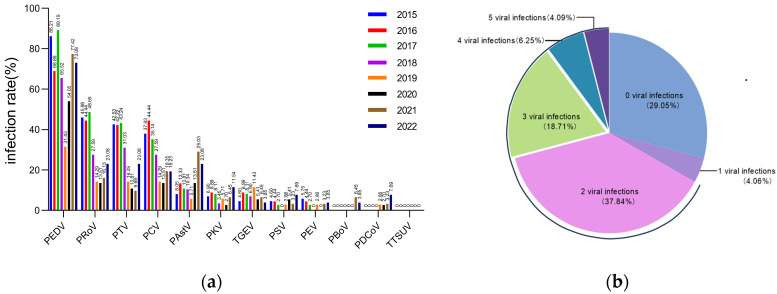
The positivity rates for selected enteroviruses in piglet diarrheal samples from 2015 to 2022. (**a**) Positivity rates for 12 designated enteroviruses in 327 diarrhea samples taken from piglets between 2015 and 2022. (**b**) Mixed infection patterns of 12 selected enteroviruses in 327 diarrhea samples between 2015 and 2022.

**Figure 2 animals-14-02168-f002:**
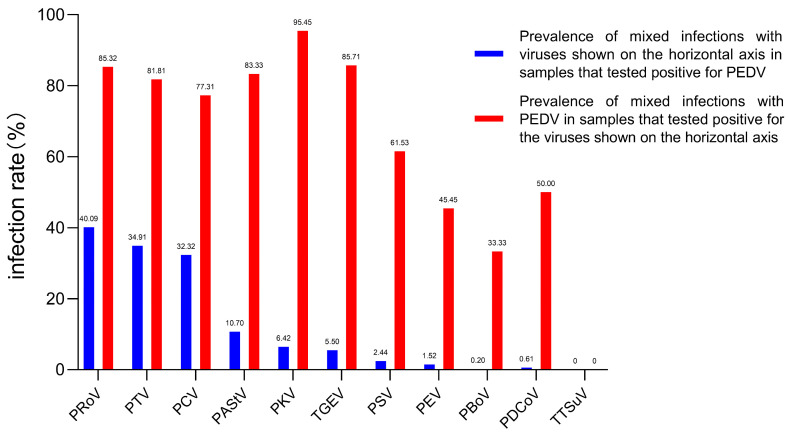
Co-infection of PEDV with selected enteroviruses in diarrhea samples from 2015 to 2022. The blue bar represents the prevalence of PEDV in samples that tested positive for PEDV and other viruses. The prevalence of mixed infection with PEDV for each of the other 11 diarrhea viruses is indicated by the red bar.

**Figure 3 animals-14-02168-f003:**
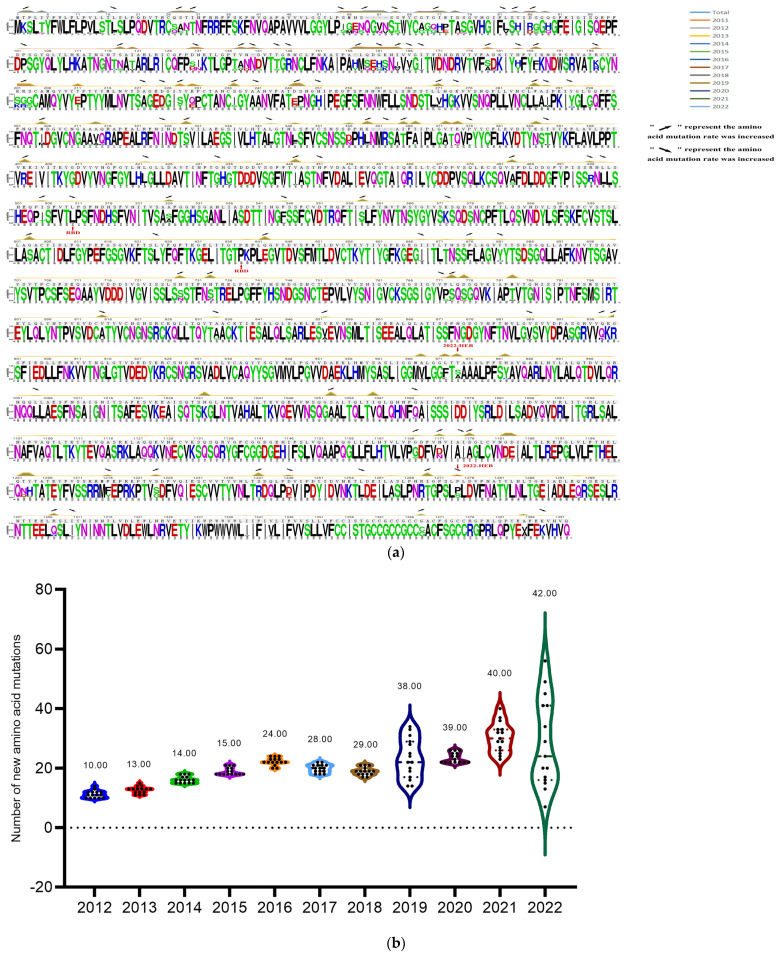
Sequence analysis of S proteins of the identified PEDV strains from 2011 to 2022. (**a**) The amino acids of 732 prevalent PEDV S genes from 2011 to 2022 in NCBI Genbank, and 155 S genes identified in this study were used to analyze the amino acid mutation of S protein of PEDV strains. The lines with different colors in the figure represent the corresponding years of the strains, and the rising positions of the lines indicate the amino acid mutation positions. Rising arrows indicate an increase in the probability of co-mutation, while falling arrows indicate a decrease in the probability of co-mutation. The gray underlined letters are the CV777 sequences; the colored letters illustrate the shared sequences of the representative strains in all the years, and the reduced letters depict the mutation patterns. (**b**) Annual variation statistics of amino acid mutations in S proteins of all selected PEDV strains (*n* = 732 + 155) prevalent worldwide from 2012 to 2022. The vertical coordinate indicates the number of mutations, the horizontal coordinate is the year of PEDV prevalence, and the dots shown in the violin plot indicate the number of mutations in the S protein of PEDV strains prevalent in each year.

**Figure 4 animals-14-02168-f004:**
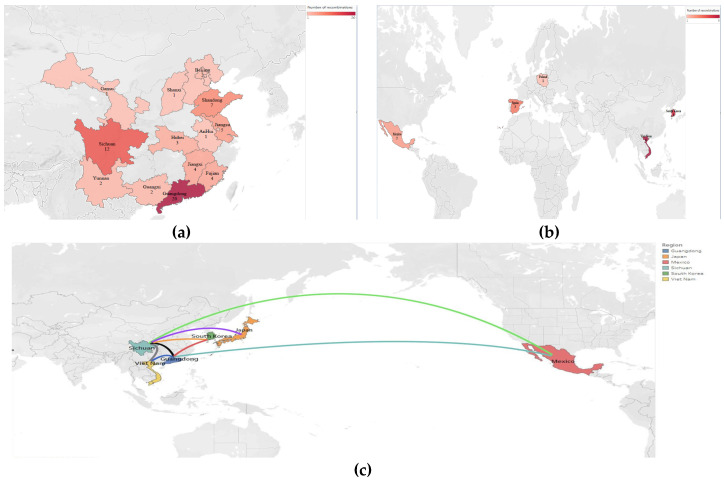
Distribution of recombination events of prevalent strains in China and other countries. (**a**) Distribution of recombination events in PEDV strains prevalent in non-China regions. (**b**) Recombination event distribution of prevalent PEDV strains in Chinese geographic regions. (**c**) Association of recombination events of PEDV strains prevalent in China with those in other countries.

**Figure 5 animals-14-02168-f005:**
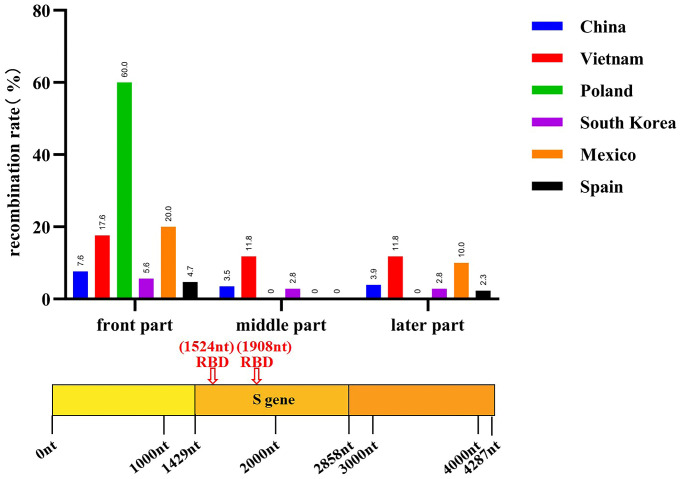
Recombination positions of global PEDV prevalent strains (*n* = 732 + 155) from 2011 to 2022.

**Figure 6 animals-14-02168-f006:**
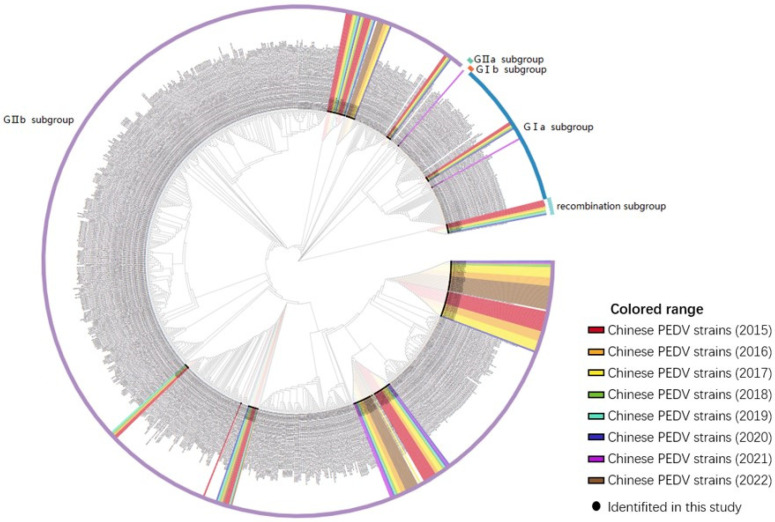
S gene-based phylogenetic analysis of northeast China endemic PEDV strains (*n* = 155). The colors indicate the year in which the strains were identified, and the lines in the outer circle of the evolutionary tree represent the subtypes of the strains.

**Figure 7 animals-14-02168-f007:**
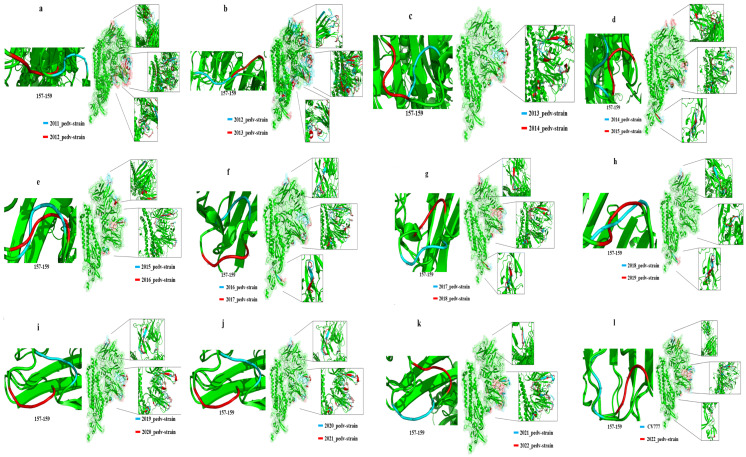
Year-on-year comparison of the S protein modeling differences between 2011 and 2022 prevalent PEDV strains. (**a**) 2011 and 2012–PEDV strains; (**b**) 2012 and 2013–PEDV strains; (**c**) 2013 and 2014–PEDV strains; (**d**) 2014 and 2015–PEDV strains; (**e**) 2015 and 2016–PEDV strains; (**f**) 2016 and 2017–PEDV strains; (**g**) 2017 and 2018–PEDV strains; (**h**) 2018 and 2019–PEDV strains; (**i**) 2019 and 2020–PEDV strains; (**j**) 2020 and 2021–PEDV strains; (**k**) 2021 and 2022–PEDV strains;(**l**) CV777 and 2022–PEDV strains.

**Table 1 animals-14-02168-t001:** New mutations in the S protein of representative PEDV strains between 2011 and 2022.

Year	New Amino Acid Mutations
2012–2013	Y293H, Q317P, T373I, I505T, S576T, K639I, T833I, V852A, S974A, P1274S
2013–2014	L510P, Q898K
2014–2015	V556A, R145L
2015–2016	L59S, T365I, T496R, P504S, I505T, P772S, V852A, S974A
2016–2017	I325T, S558T, A834S, V1343A
2017–2018	P1274S
2018–2019	L271V, A288T, M289T, N451T, S526T, V886A, S974A, A1095S, A1263S
2019–2020	G296V
2020–2021	D327A
2021–2022	S974A

**Table 2 animals-14-02168-t002:** The year-on-year range of structural residues of protein S of the representative PEDV strains.

Year of Comparison	NTD (aa)	RBD (aa)
CV777-HEB (2022)	55–72, 85–117, 137–140, 156–162, 188, 199–210, 211–231	528–536
2021-HEB (2022)	55–64, 114–117, 132–141, 151–164, 184, 199–201, 209–213	563–565
2020–2021	55–59, 65–68, 111–113, 132–135, 150–160, 184, 195–197, 206–218	520–529, 556–557
2019–2020	157–160	-
2018–2019	157–160	520–553
2017–2018	57–64, 68–72, 115–117, 138–141, 156–164, 199–201, 210–225	524–530
2016–2017	55–60, 65–68, 111–113, 132–135, 151–160, 195–197, 205–219	520–527
2015–2016	56–61, 112–114, 134–135, 207–211	526
2014–2015	56–61, 112–114, 134–135, 207–211	526
2013–2014	52–56, 65–68, 111–113, 126–135, 153–159, 196–198, 206–220	520–529, 556–557
2012–2013	38–40, 43–51, 56–76, 84–88, 99–102, 115–117, 133–147, 155–161, 177–179, 184–185, 197–202, 209–217	522–532, 560–567

## Data Availability

The authors confirm that the data supporting the findings of this study are available within the article and the reference genome sequence of the endemic PEDV strains are openly available in NCBI.

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
