# Peer review of "Investigation of Transmission and Evolution of PEDV Variants and Co-Infections in Northeast China from 2011 to 2022"

_animals, 2024, doi:10.3390/ani14152168_

Round 1

Reviewer 1 Report

Comments and Suggestions for Authors

The title of this study is "Investigation of Transmission and Evolution of PEDV Variants and Co-Infections in Northern China over Past Decades". "The samples in this study come from four provinces in Northern China, which cannot represent the whole Northern region, but only some regions, and the samples collected in each region are too small to draw a conclusion that represents the whole, which is rather one-sided and misleading, and cannot represent the whole situation, so the title of the thesis is suggested to be revised.

Line31 to Line36: Please provide literature to support this section.

Line46 toLine50: Please provide literature to support this section.

Reference [7] "Virological and molecular characterization of a mammalian orthoreovirus type 3 strain isolated from a dog in Italy " does not match what you have described, please verify if it is appropriate to cite this document.

Line63 to Line77: both describe the genome structure of PEDV and should be combined into one paragraph.

Line68: "[12,13,14,15]" formatting error

Line78 to Line81: please provide literature to support this section

Line83 and Line84: references [24] are duplicated and can be combined into one

Line91 to Line95: Please provide literature to support this section.

Line97 to Line99: Please provide literature to support this section.

Line222 to Line225: the description is missing from Figure 3b.

"Figure 5": what is the basis for dividing the S gene into three parts? Label the specific loci on the figure in detail.

Line366 and Line370: reference [45] is duplicated and can be combined into one.

Line465 and Line467: duplicate of reference [44], can be combined into one.

There are too many textual descriptions in the results section, so streamlining is recommended, and some of the results are suggested to be represented in tables.

Comments on the Quality of English Language

Please make appropriate language changes.

Author Response

Thank you very much for the reviewing process of our manuscript entitled “Investigation of transmission and evolution of PEDV variants and co-infections in Northern China over past decades” (Manuscript Number: 3088549). We deeply value the reviewer's meticulousness, conscientious feedback, and constructive suggestions. All of these comments are immensely valuable and provide significant guidance for refining and enhancing our paper, as well as steering our research in the right direction. We sincerely appreciate the reviewer's dedicated efforts and earnestly hope that the revisions meet with approval.

We have revised the manuscript accordingly, with yellow highlighting in the revision section of the paper and blue highlighting in the image and table revision section, and detailed point-by-point revisions are listed below:

(We have also uploaded a Word version of the reply for you to view, as the content of the reply we have copied into the text box does not display an image.)

Reply to reviewer#1

  1. Line31 to Line36: Please provide literature to support this section.

ReplyWe have added supporting references [1], [2], [3], and [4] in Line 35-42 as follows:

[1]     Oldham J. Pig Farming (Suppl). 1972;10.

[2]     Duy DT, Toan NT, Puranaveja S, et al. Genetic Characterization of Porcine Epidemic Diarrhea Virus (PEDV) Isolates from Southern Vietnam during 2009-2010 Outbreaks. Thai Journal of Veterinary Medicine. 2011;41(1):55-64.

[3]     Takahashi K, Okada K, Ohshima K. An outbreak of swine diarrhea of a new-type associated with coronavirus-like particles in Japan. Nihon Juigaku Zasshi. 1983;45(6):829-832.

[4]     Sueyoshi M, Tsuda T, Yamazaki K, et al. An immunohistochemical investigation of porcine epidemic diarrhoea. J Comp Pathol. 1995;113(1):59-67.

  1. Line46 toLine50: Please provide literature to support this section.

ReplyWe have added supporting reference [8] in Line 51-57 as follows:

[8]     Su M, Li C, Qi S, et al. A molecular epidemiological investigation of PEDV in China: Characterization of co-infection and genetic diversity of S1-based genes. Transbound Emerg Dis. 2020;67(3):1129-1140.

  1. Reference [7] "Virological and molecular characterization of a mammalian orthoreovirus type 3 strain isolated from a dog in Italy " does not match what you have described, please verify if it is appropriate to cite this document.

ReplyWe have noted this issue and changed it to reference [11] in line 59.

[11] Zhang Q, Hu R, Tang X, et al. Occurrence and investigation of enteric viral infections in pigs with diarrhea in China. Arch Virol. 2013;158(8):1631-1636.

  1. Line63 to Line77: both describe the genome structure of PEDV and should be combined into one paragraph.

ReplyWe have rewritten the paragraph in lines 69-76. The paragraph is now amended as follows: “PEDV belongs to the Alphacoronavirus genus in the subfamily of Coronaviridae. It is a non-segmented, capsid, single-stranded RNA virus with a positive-stranded genome approximately 28000 nucleotides [6]. The genome contains seven open reading frames and non-coding regions situated at the 5′ and 3′ ends. Two-thirds of the 5′ end of the PEDV genome encodes the polyproteins ORF1a and ORF1b, which play important roles in viral replication and suppression of host innate immunity [16-19]. The sequence's 3′ end of the sequence encodes several proteins, including the spike protein (S), envelope protein (E), membrane protein (M), nucleocapsid protein (N), and an accessory protein to ORF3 [18].”

  1. Line68: "[12,13,14,15]" formatting error

ReplyIn order to focus the article, we have optimized the original text in lines 74-76 by simplifying the description of the N, E, M and ORF3 genes. Consequently, literature 12, 13, 14, and 15 on the functional elaboration of these genes were removed.

  1. Line78 to Line81: please provide literature to support this section

ReplyWe have added supporting reference [28] in Line 77-80 as follows:

[28]   Millet JK, Whittaker GR. Host cell proteases: Critical determinants of coronavirus tropism and pathogenesis. Virus Res. 2015; 202:120-134.

  1. Line83 and Line84: references [24] are duplicated and can be combined into one

ReplyWe have revised the content in lines 80-83 as suggested.

The paragraph is now amended as follows:“Recent studies have shown that the S1 region of the S protein plays a key role in determining the pathogenicity of PEDV. This region is also responsible for facilitating the attachment of viral particles to cell surface receptors [29].”

  1. Line91 to Line95: Please provide literature to support this section.

ReplyWe have added supporting references [31] and [32] in Line 90-94 as follows:

[31]   Deng F, Ye G, Liu Q, et al. Identification and Comparison of Receptor Binding Characteristics of the Spike Protein of Two Porcine Epidemic Diarrhea Virus Strains. Viruses. 2016;8(3):55.

[32.]  Heald-Sargent T, Gallagher T. Ready, set, fuse! The coronavirus spike protein and acquisition of fusion competence. Viruses. 2012;4(4):557-580.

  1. Line97 to Line99: Please provide literature to support this section.

ReplyWe have added supporting reference [36] in Line 102-104 as follows:

[36] Wang L, Byrum B, Zhang Y. New variant of porcine epidemic diarrhea virus, United States, 2014. Emerg Infect Dis. 2014;20(5):917-919.

  1. Line222 to Line225: the description is missing from Figure 3b.

ReplyWe have modified the figure note to Figure 3b by adding“Points shown in the violin plot are labelled as the number of mutations in the S protein of the prevalent PEDV strains detected in each year.”in lines 251-254.

  1. "Figure 5": what is the basis for dividing the S gene into three parts? Label the specific loci on the figure in detail.

ReplyWe divided the S gene into three equal parts according to the length of the gene, so that the recombination differences between the strains prevalent in each region could be shown more clearly. Also, we have modified Figure 5 to include the details of the special localization points. The new Figure 5 is as follows:

Figure 5. Statistics on the recombination positions of prevalent PEDV strains from 2011 to 2022.

  1. Line366 and Line370: reference [45] is duplicated and can be combined into one.

ReplyWe have revised the content in lines 349-354 as suggested.

  1. Line465 and Line467: duplicate of reference [44], can be combined into one.

ReplyWe have combined the reference and revised the content in lines 466-470 as suggested.

  1. There are too many textual descriptions in the results section, so streamlining is recommended, and some of the results are suggested to be represented in tables.

ReplyIn order to facilitate comprehension, the descriptions in Section 3.2 were presented in graphical form (Table 1) at line 241-242 and lines 255, whereby the year-to-year increase in mutation sites in the PEDV S protein was illustrated. Similarly, in Section 3.5, the annual changes in new PEDV S protein sites were presented in Table 2. at lines 302-304 and lines 329.

Table 1. Statistics of new mutations in PEDV S protein year by year.Year

Year

New amino acid mutations

2012-2013

Y293H、Q317P、T373I、I505T、S576T、K639I、T833I、V852A、S974A 、 P1274S

2013-2014

L510P 、Q898K

2014-2015

V556A 、 R145L

2015-2016

L59S、T365I、T496R、P504S、I505T、P772S、V852A 、S974A

2016-2017

I325T、S558T、A834S、V1343A

2017-2018

P1274S

2018-2019

L271V、A288T、M289T、N451T、S526T、V886A、S974A、A1095S、A1263S

2019-2020

G296V

2020-2021

D327A

2021-2022

S974A

Table 2. Statistics on year-to-year changes in PEDV S protein additions

Year of comparison

NTDaa

RBDaa

CV777-HEB(2022)

55-72、85-117、137-140、156-162、188、199-210、211-231

528-536

2021-HEB(2022)

55-64、114-117、132-141、151-164、184、199-201、209-213

563-565

2020-2021

55-59、65-68、111-113、132-135、150-160、184、195-197、206-218

520-529、556-557

2019-2020

157-160

-

2018-2019

157-160

520-553

2017-2018

57-64、68-72、115-117、138-141、156-164、199-201、210-225

524-530

2016-2017

55-60、65-68、111-113、132-135、151-160、195-197、205-219

520-527

2015-2016

56-61、112-114、134-135、207-211

526

2014-2015

56-61、112-114、134-135、207-211

526

2013-2014

52-56、65-68、111-113、126-135、153-159、196-198、206-220

520-529、556-557

2012-2013

38-40、43-51、56-76、84-88、99-102、115-117、133-147、155-161、177-179、184-185、197-202、209-217

522-532、560-567

Reviewer 2 Report

Comments and Suggestions for Authors

The authors investigated the pathogenesis and genetic evolution of porcine epidemic diarrhea virus (PEDV) in northern China. They show that PEDV is commonly found together with eleven other enteroviruses. The PEDV analyzed in the study belonged to the Chinese endemic strains and showed genetic diversity. Mutations in the S gene were found, which alter the protein structure. The S gene is divided into S1 and S2, S1 into a N-terminal domain (NTD) and a C-terminal domain (CTD). S1-NTD is associated with virulence, the S1-CTD contains the receptor binding site and binds to the receptor, the porcine aminopeptidase N (pAPN). Sugar residues serve as co-receptors. Furthermore, recombinations were found which were similar to recombinations described in other countries.

They highlight that GII strains are undergoing more rapid evolution, certainly because conventional vaccines exist, which provide superior protection against GI subtypes.

Question: The authors claim that the PEDV found belong to the Chinese endemic strains, on the other hand they associate recombinations with export and import in/from other countries.

The problem with the manuscript is the fact that only very small number of pigs were infected with PEDV alone and in most cases the animals were co-infected with other viruses, up to eleven others including swine diarrhea, including TGEV, Porcine Teschovirus (PTV), Porcine Sapelovirus (PSV), Porcine Enterovirus (PEV), Porcine Rotavirus (PRoV), Porcine Astrovirus (PAstV), Porcine Circovirus (PCV), Porcine Bocavirus (PBoV), Porcine Kobuvirus (PKV)Torque Teno Sus virus (TTSuV-2), and Porcine Delta coronavirus (PDCoV).

Therefore, the statement of the author that “Recent data suggests that PEDV is the primary cause of viral diarrheal disease in pigs throughout China.” is unclear. 67% of the sample from diarrheal diseases contained at least two viruses, PEDV alone was found in only 2.75% of the samples.

It would be important to show that all diarrheal diseases induced by these viruses alone or together are clinically identical or different. I guess that there were differences in severity and duration of the diarrheal diseases in dependence of the number of infecting viruses and the combination of viruses. This information will be crucial. Furthermore, did the authors found animals which are positive for PEDV and had no disease?

Line 41: What means post-immunized, please explain.

Line 472: The authors underline that co-infections with PRoV and PTV were associated with clinical signs of PEDV-induced diarrhea. Again my questions from the beginning: are the clinical signs of the other cases, e.g., co-infections with other viruses or with PEDV alone different??

Figure 1b. 29% of the diarrheal disease are marked as 0 viral infections, what is the cause of these diseases?

Figure 2: It is unclear what is shown, the authors should explain it in the legends in detail

Figure 3a is of very bad quality, please mark the receptor binding domain (RBD) in the sequence

Figure 5 please mark the RBD in the C-terminal part of S

Minor

Lines 58-59: in this sentence information is missing

Line 94: what means pAPN, please explain that this is the receptor and that there may sugar residues as co-receptors

Lines 123-124: it should be: RNA extraction and cDNA synthesis were followed the protocols described by Wang et al. [32], DNA extraction was conducted based on the protocol by Qi et al [33].

Comments on the Quality of English Language

fine

Author Response

Thank you very much for the reviewing process of our manuscript entitled “Investigation of transmission and evolution of PEDV variants and co-infections in Northern China over past decades” (Manuscript Number: 3088549). We deeply value the reviewer's meticulousness, conscientious feedback, and constructive suggestions. All of these comments are immensely valuable and provide significant guidance for refining and enhancing our paper, as well as steering our research in the right direction. We sincerely appreciate the reviewer's dedicated efforts and earnestly hope that the revisions meet with approval.

We have revised the manuscript accordingly, with yellow highlighting in the revision section of the paper and blue highlighting in the image and table revision section, and detailed point-by-point revisions are listed below:

(We have also uploaded a Word version of the reply for you to view, as the content of the reply we have copied into the text box does not display an image.)

Reply to reviewer#2

  1. Question: The authors claim that the PEDV found belong to the Chinese endemic strains, on the other hand they associate recombination with export and import in/from other countries.

ReplyThe PEDV strains we identified were consistent with the sequence characteristics of the Chinese endemic strains, and recombination analyses revealed recombination of endemic and non-Chinese strains in the Chinese region, and a correlation of transmission between endemic strains in these two regions.

  1. The problem with the manuscript is the fact that only very small number of pigs were infected with PEDV alone and in most cases the animals were co-infected with other viruses, up to eleven others including swine diarrhea, including TGEV, Porcine Teschovirus (PTV), Porcine Sapelovirus (PSV), Porcine Enterovirus (PEV), Porcine Rotavirus (PRoV), Porcine Astrovirus (PAstV), Porcine Circovirus (PCV), Porcine Bocavirus (PBoV), Porcine Kobuvirus (PKV)Torque Teno Sus virus Ⅱ (TTSuV-2), and Porcine Delta coronavirus (PDCoV).

Therefore, the statement of the author that “Recent data suggests that PEDV is the primary cause of viral diarrheal disease in pigs throughout China.” is unclear. 67% of the sample from diarrheal diseases contained at least two viruses, PEDV alone was found in only 2.75% of the samples.

ReplyAmong the samples we tested, 70.9% (232/327) were positive for PEDV, which had the highest prevalence. Although the test results showed that most of the PEDV infections were mixed with other viruses, the percentage was still the highest. Combined with the article cited in line 57, " Recent data suggests that PEDV is the primary cause of viral diarrheal disease in pigs throughout China [8]", we determined that PEDV is the main pathogen responsible for viral diarrhea in Chinese piglets.

[8]     Su M, Li C, Qi S, et al. A molecular epidemiological investigation of PEDV in China: Characterization of co-infection and genetic diversity of S1-based genes. Transbound Emerg Dis. 2020;67(3):1129-1140.

  1. It would be important to show that all diarrheal diseases induced by these viruses alone or together are clinically identical or different. I guess that there were differences in severity and duration of the diarrheal diseases in dependence of the number of infecting viruses and the combination of viruses. This information will be crucial. Furthermore, did the authors found animals which are positive for PEDV and had no disease?

Reply: The severity of diarrheal symptoms is a certain associated with the virus type involved in the co-infection according to the available research(derived from Jung and Saeng-Chuto et al). In our investigation, we only did genetic testing on samples of piglet diarrhea, and did not record or observe the clinical condition of the affected animals; therefore, we do not have data related to the severity of diarrhea after mixed infections. In addition, since all the samples collected in the survey were diarrhea samples, we did not examine samples without diarrhea symptoms, and therefore have no relevant data.

  1. Jung K, Kang B K, Lee C S, et al.Impact of porcine group A rotavirus co-infection on porcine epidemic diarrhea virus pathogenicity in piglets[J].Research in Veterinary Science, 2008, 84(3):502-506.DOI:10.1016/j.rvsc.2007.07.004.
  2. Saeng-Chuto K, Madapong A, Kaeoket K, et al.Coinfection of porcine deltacoronavirus and porcine epidemic diarrhea virus increases disease severity, cell trophism and earlier upregulation of IFN-α and IL12[J].Scientific Reports, 2021, 11(1).DOI:10.1038/s41598-021-82738-8.

4 Line 41: What means post-immunized, please explain.

ReplyWe extend our apologies for any distress caused by the original statement. Consequently, we have rewritten the original sentence to convey the following in line 43-47: “Since then, the virus has spread throughout Asia and can be controlled with a vaccine [5]. But after 2006, new strains of PEDV began to emerge and proliferate in pigs that had been immunised with the original PEDV vaccine [6]. This phenomenon suggests that the original vaccine had lost its efficacy against the new strain, and that new strains had begun to circulate.”

[5]  Takahashi K, Okada K, Ohshima K. An outbreak of swine diarrhea of a new-type associated with coronavirus-like particles in Japan. Nihon juigaku zasshi The Japanese journal of veterinary science. 1983;45(6):829-832.

[6]  Chen J, Wang C, Shi H, et al. Molecular epidemiology of porcine epidemic diarrhea virus in China. Archives of virology. 2010;155(9):1471-1476.

  1. Line 472: The authors underline that co-infections with PRoV and PTV were associated with clinical signs of PEDV-induced diarrhea. Again my questions from the beginning: are the clinical signs of the other cases, e.g., co-infections with other viruses or with PEDV alone different??

ReplyThe severity of diarrheal symptoms is a certain associated with the virus type involved in the co-infection according to the available research. We have rewritten it as " In conclusion, our study provides evidence that PEDV co-infects with 11 enteroviruses. " in line 475.In this survey, we only collected and genetically tested diarrheal stool samples; we did not collect and record data on clinical symptoms, and therefore do not have data related to clinical symptoms.

  1. Figure 1b. 29% of the diarrheal disease are marked as 0 viral infections, what is the cause of these diseases?

ReplyWe have added a related discussion of " It is noteworthy that no viral infection was detected in 29.05% of the samples we tested. Due to the multitude of clinical causes that can lead to swine diarrhea, we hypothesize that the samples with no detectable viral infections may be the result of bacterial infections or may be the result of stress during the feeding process. " in line 344-347.

  1. Figure 2: It is unclear what is shown, the authors should explain it in the legends in detail

ReplyWe added the following to the graphical note to Figure 2 in line 206-208 " The blue bar represents the prevalence of PEDV in samples that tested positive for PEDV and other viruses. The prevalence of mixed infection with PEDV for each of the other 11 diarrhea viruses is indicated by the red bar."

  1. Figure 3a is of very bad quality, please mark the receptor binding domain (RBD) in the sequence

Reply We modified the image and labeled the location of the receptor binding domain (RBD) in Figure 3a, and the modified image is as follows:

Figure 3. Sequence analysis of the S proteins of the identified PEDV strains. (a) Analysis of the variability of the S protein in identified PEDV strains from 2011 to 2022.

  1. Figure 5 please mark the RBD in the C-terminal part of S

Reply We modified Figure 5 to mark the location of the RBD, and the modified image is as follows:

Minor

  1. Lines 58-59: in this sentence information is missing

ReplyWe have revised the original text to read " In addition to PEDV, several other porcine enteric viruses are responsible for swine diarrhea, including Transmissible gastroenteritis virus (TGEV), Porcine Teschovirus (PTV), Porcine Sapelovirus (PSV), Porcine Enterovirus (PEV), Porcine Rotavirus (PRoV), Porcine Astrovirus (PAstV), Porcine Circovirus (PCV), Porcine Bocavirus (PBoV), Porcine Kobuvirus (PKV),Torque Teno Sus virus (TTSuV-2), and Porcine Delta coronavirus (PDCoV). Recent data suggests that PEDV is the primary cause of viral diarrheal disease in pigs throughout China [8]." in line 51-57 to add the following reference:

[8]     Su M, Li C, Qi S, et al. A molecular epidemiological investigation of PEDV in China: Characterization of co-infection and genetic diversity of S1-based genes. Transbound Emerg Dis. 2020;67(3):1129-1140.

  1. Line 94: what means pAPN, please explain that this is the receptor and that there may sugar residues as co-receptors

ReplyWe apologize for the lack of clarity in the previous description, so we have rewritten the description of this section in line 90-95 of the article as " The S1 region of all coronaviruses contains the receptor binding domain (RBD), which recognizes cellular receptors to facilitate viral entry [32]. The classical weak PEDV strain CV777 and the mutant strain GHGD-01 do not differ significantly in the ability to bind porcine aminopeptidase N (pAPN), the primary receptor of PEDV. However, GHGD-01 is more capable of recognising glycosylated proteins [31]. This may explain the difference in virulence between the two strains."

  1. Lines 123-124: it should be: RNA extraction and cDNA synthesis were followed the protocols described by Wang et al. [32], DNA extraction was conducted based on the protocol by Qi et al [33].

Reply We sincerely appreciate your suggestion, and we have revised lines 134-135 of the article as " RNA extraction and cDNA synthesis were followed the protocols described by Wang et al[43]. DNA extraction was conducted based on the protocol by Qi et al [44]."

[43]   Wang E, Guo D, Li C, et al. Molecular Characterization of the ORF3 and S1 Genes of Porcine Epidemic Diarrhea Virus Non S-INDEL Strains in Seven Regions of China, 2015. PloS one. 2016;11(8):e0160561.

[44]   Qi S, Su M, Guo D, et al. Molecular detection and phylogenetic analysis of porcine circovirus type 3 in 21 Provinces of China during 2015-2017. Transboundary and emerging diseases. 2019;66(2):1004-1015.

Reviewer 3 Report

Comments and Suggestions for Authors

The manuscript showed the results of the epidemiological survey of pigs in northern China considering PEDV co-infection from 2015 to 2022 and the genetic evolution of S gene from 2011 to 2022. Initially, co-infections of PEDV-infected pigs were analyzed concerning 11 other enteroviruses. Then S gene sequencing was evaluated and compared with GenBank data for recombination and phylogenetic analyses. The authors did not correlate the S gene sequencing samples with co-infection results done in the first part of the manuscript. Two distinct mutations in 2022 HEB strain were identified showing genetic evolution. The meaning and importance of these results in the virus evolution and pathogenicity were not explained in the manuscript. The novelty of the data was not explored, there was a description of survey results and comparisons with GenBack data.

The manuscript sounded like a communication article, did the authors think another category for it?

The title mentioned “… over Past Decades” and the term is inappropriate because the authors presented epidemiological survey data from 2015 to 2022 and the mutation of S gene from 2011 to 2022.

- “Introduction section” must be improved focusing on fields cited in the manuscript. Reduce the explanation of the virus structure and explore the types and classification of PEDV strains found until now.

It is not clear if there is differences in the PEDV infection concerning epidemiological data among diverse regions of China. What is the importance of presenting a specific study of the northern region?

Lanes 91-94: “The classical weak PEDV strain CV777 and the mutant strain GHGD-01 do not differ significantly in pAPN binding capacity.” Explain the meaning and the importance of this sentence.

- “Results” section: the main data obtained in the manuscript should be summarized in the table to facilitate the understanding of the readers.

- "Discussion section" should correlate the results obtained in the manuscript with published articles. There is the citation of published articles but a poor comparison with the manuscript results. And the citation of recently published articles in the field could improve the discussion quality.

Author Response

Thank you very much for the reviewing process of our manuscript entitled “Investigation of transmission and evolution of PEDV variants and co-infections in Northern China over past decades” (Manuscript Number: 3088549). We deeply value the reviewer's meticulousness, conscientious feedback, and constructive suggestions. All of these comments are immensely valuable and provide significant guidance for refining and enhancing our paper, as well as steering our research in the right direction. We sincerely appreciate the reviewer's dedicated efforts and earnestly hope that the revisions meet with approval.

We have revised the manuscript accordingly, with yellow highlighting in the revision section of the paper and blue highlighting in the image and table revision section, and detailed point-by-point revisions are listed below:

Reply to reviewer#3

  1. The manuscript sounded like a communication article, did the authors think another category for it?

ReplyWe extend our apologies for any confusion that may have arisen. In this article, we initially collected a substantial number of faecal samples from piglets exhibiting diarrhea, subjected the samples to analysis of viral co-infections, and identified PEDV as the most prevalent virus. The S gene mutation pattern analysis, phylogenetic analysis, and protein homology modelling analysis were subsequently performed to ascertain the prevalence and evolving patterns of PEDV in China. Accordingly, this article was submitted as a research manuscript.

  1. The title mentioned “… over Past Decades” and the term is inappropriate because the authors presented epidemiological survey data from 2015 to 2022 and the mutation of S gene from 2011 to 2022.

Reply: Thanks to your suggestion, we have revised the title to be more specific as “Investigation of transmission and evolution of PEDV variants and co-infections in Northeast China from 2011 to 2022”.

  1. “Introduction section” must be improved focusing on fields cited in the manuscript. Reduce the explanation of the virus structure and explore the types and classification of PEDV strains found until now.

ReplyWe have revised this section in lines 97-104 to briefly describe genes other than the S gene. An extended description of the genotypes of PEDV strains that are prevalent at this stage is also provided. This part is now modified to“Phylogenetic analysis of S sequences classified PEDV into two genotypes, G1 and G2. It is notable that the classical strain CV777, which was prevalent in the early years, belongs to the G1 subgroup, whereas globally prevalent of PEDV strains after 2010 are dominated by strains in the G2 subgroup [34]. Moreover, the majority of PEDVs identified in China at this stage are of  the G2 subtype [35]. In 2013, attenuated strains of PEDV, namely S-INDEL and OH851, were initially identified in the USA, and have subsequently been reported in other countries [36].”

[34]   Chen J, Liu X, Shi D, et al. Detection and molecular diversity of spike gene of porcine epidemic diarrhea virus in China. Viruses. 2013;5(10):2601-2613.

[35] Wang J, Zhao P, Guo L, et al. Porcine epidemic diarrhea virus variants with high pathogenicity, China. Emerg Infect Dis. 2013;19(12):2048-2049.

[36]   Wang L, Byrum B, Zhang Y. New variant of porcine epidemic diarrhea virus, United States, 2014. Emerg Infect Dis. 2014;20(5):917-919.

  1. It is not clear if there is differences in the PEDV infection concerning epidemiological data among diverse regions of China. What is the importance of presenting a specific study of the northern region?

ReplyChina is a large region with great differences in climate, temperature and precipitation between the northern and southern parts of the country. Thus, the characteristics of the PEDV epidemic in the northeast may be specific. To further illustrate the significance of this study, we have explained it in line 115-118,as follows:“The Northeast region of China is also a major swine breeding area, but it differs greatly in terms of climate, temperature and precipitation compared to the Southern part of the country. There are significantly long and cold seasons in winter, which may have an impact on the PEDV emergence, transmission and variation.”

  1. Lanes 91-94: “The classical weak PEDV strain CV777 and the mutant strain GHGD-01 do not differ significantly in pAPN binding capacity.” Explain the meaning and the importance of this sentence.

ReplyOur apologies for the previous lack of clarity, so we've rewritten the description of this section in line 90-95 of the article,as follows:“The S1 region of all coronaviruses contains the receptor binding domain (RBD), which recognizes cellular receptors to facilitate viral entry [32]. The classical weak PEDV strain CV777 and the mutant strain GHGD-01 do not differ significantly in the ability to bind porcine aminopeptidase N (pAPN), the primary receptor of PEDV. However, GHGD-01 is more capable of recognising glycosylated proteins [31]. This may explain the difference in virulence between the two strains.”

[31]   Deng F, Ye G, Liu Q, et al. Identification and Comparison of Receptor Binding Characteristics of the Spike Protein of Two Porcine Epidemic Diarrhea Virus Strains. Viruses. 2016;8(3):55.

[32]   Heald-Sargent T, Gallagher T. Ready, set, fuse! The coronavirus spike protein and acquisition of fusion competence. Viruses. 2012;4(4):557-580.

  1. “Results” section: the main data obtained in the manuscript should be summarized in the table to facilitate the understanding of the readers.

ReplyIn order to facilitate comprehension, the descriptions in Section 3.2 were presented in graphical form (Table 1) at line 241-242 and lines 255, whereby the year-to-year increase in mutation sites in the PEDV S protein was illustrated. Similarly, in Section 3.5, the annual changes in new PEDV S protein sites were presented in Table 2. at lines 302-304 and lines 329.

Table 1. Statistics of new mutations in PEDV S protein year by year.Year

Year

New amino acid mutations

2012-2013

Y293H、Q317P、T373I、I505T、S576T、K639I、T833I、V852A、S974A 、 P1274S

2013-2014

L510P 、Q898K

2014-2015

V556A 、 R145L

2015-2016

L59S、T365I、T496R、P504S、I505T、P772S、V852A 、S974A

2016-2017

I325T、S558T、A834S、V1343A

2017-2018

P1274S

2018-2019

L271V、A288T、M289T、N451T、S526T、V886A、S974A、A1095S、A1263S

2019-2020

G296V

2020-2021

D327A

2021-2022

S974A

Table 2. Statistics on year-to-year changes in PEDV S protein additions

Year of comparison

NTDaa

RBDaa

CV777-HEB(2022)

55-72、85-117、137-140、156-162、188、199-210、211-231

528-536

2021-HEB(2022)

55-64、114-117、132-141、151-164、184、199-201、209-213

563-565

2020-2021

55-59、65-68、111-113、132-135、150-160、184、195-197、206-218

520-529、556-557

2019-2020

157-160

-

2018-2019

157-160

520-553

2017-2018

57-64、68-72、115-117、138-141、156-164、199-201、210-225

524-530

2016-2017

55-60、65-68、111-113、132-135、151-160、195-197、205-219

520-527

2015-2016

56-61、112-114、134-135、207-211

526

2014-2015

56-61、112-114、134-135、207-211

526

2013-2014

52-56、65-68、111-113、126-135、153-159、196-198、206-220

520-529、556-557

2012-2013

38-40、43-51、56-76、84-88、99-102、115-117、133-147、155-161、177-179、184-185、197-202、209-217

522-532、560-567

  1. "Discussion section" should correlate the results obtained in the manuscript with published articles. There is the citation of published articles but a poor comparison with the manuscript results. And the citation of recently published articles in the field could improve the discussion quality.

ReplyIn this context, we refer to the literature cited in [54] and [62] and have modified the discussion in lines 403-417. This section presents the findings of recent studies on the characterisation of mutations and mutational trends in S proteins of PEDV. The results are then compared and discussed with those of ourstudy, with the conclusions presented in lines 403-417.Modified as follows: “These results corroborate the findings of Chen et al [62], who observed a steady increase in the number of new mutation sites in the dominant PEDV strains between 2011 and 2022, accompanied by a stabilization of the structural mutations in the viral NTD and RBD. These structures may be potential key sites for the virus. Zhang et al. observed that mutations in the PEDV S protein resulted in an increase in the α-helical structure at residues 55-64 and the replacement of the β-turn by an α-helix at residues 157-164, which altered the virulence, pathogenicity, and immunogenicity of the virus and led to the evolution of the virus from subgroup G1 to subgroup G2. Residues 55-64 are specific for recognizing G1 strains, while residues 157-164 exhibited specificity in terms of immunogenicity [54]. Our study revealed that the region of PEDV S protein expanded to residues 55-64, 113-117, 133, 157-159, 198 and 210 between 2011 and 2022. Furthermore, fluctuations in these residues on an annual basis may impede the ability to discern evolutionary characteristics of PEDV. The constant changes in residues 55-64 may indicate that the newly prevalent strains in recent years are diverging from the immunological profile of the classical strain (G1).”

[54]   Zhang H, Zou C, Peng O, et al. Global Dynamics of Porcine Enteric Coronavirus PEDV Epidemiology, Evolution, and Transmission. Molecular biology and evolution. 2023;40(3)

[62]   Chen P, Wang K, Hou Y, et al. Genetic evolution analysis and pathogenicity assessment of porcine epidemic diarrhea virus strains circulating in part of China during 2011-2017. Infection, genetics and evolution : journal of molecular epidemiology and evolutionary genetics in infectious diseases. 2019;69:153-165.

Reviewer 4 Report

Comments and Suggestions for Authors

The manuscript by Zhao and colleagues addresses an important problem of transmission of porcine epidemic diarrhea virus (PEDV). The authors collected and analyzed information concerning the emergence of PEDV variations and mutations in S protein. The manuscript contains a number of interesting observations concerning the statistics of protein alterations and predicted results of mutations. Besides, the authors studied geographical patterns of PEDV transmission and recombination. The design of study is straightforward, the bioinformatic methods used appear to be relevant. Presented data look convincing, and the illustrations are qualitative and contribute understanding. The obtained results are well discussed and the discussion part is comprehensive. Overall, the paper can be considered for publication in the journal Animals. However, the manuscript could benefit from addressing some minor issues.

  1. Introduction - Please check all the taxa names and correct classification using the official ICTV site (https://ictv.global/taxonomy)

  2. Please use italics in the names of viral taxa where needed

  3. Section 2.5. - please indicate software versions including iTOL and Clustal

  4. Section 2.5 - please indicate the date or versions of databases used

  5. Section 3.4 - Phylogenetic analysis would benefit from comparing the trees constructed using also separate genes. You could talk about recombinations more reasoned

  6. Again, I do not understand which results indicated recombinations.

  7. Section 3.5 - Perhaps, the table containing a list of alterations would simplify reading.

  8. Figure 7 - I really do not understand why you used homology modeling while AlphaFold is convenient, simple and gives much more precise results.

  9. Line 412 - please replace “beta” with a greek letter

Author Response

Thank you very much for the reviewing process of our manuscript entitled “Investigation of transmission and evolution of PEDV variants and co-infections in Northern China over past decades” (Manuscript Number: 3088549). We deeply value the reviewer's meticulousness, conscientious feedback, and constructive suggestions. All of these comments are immensely valuable and provide significant guidance for refining and enhancing our paper, as well as steering our research in the right direction. We sincerely appreciate the reviewer's dedicated efforts and earnestly hope that the revisions meet with approval.

We have revised the manuscript accordingly, with yellow highlighting in the revision section of the paper and blue highlighting in the image and table revision section, and detailed point-by-point revisions are listed below:

Reply to reviewer#4

  1. Introduction - Please check all the taxa names and correct classification using the official ICTV site (https://ictv.global/taxonomy)

ReplyWe have made a related change in line 69. Modified as follows:“PEDV belongs to the Alphacoronavirus genus in the subfamily of Coronaviridae. It is a non-segmented, capsid, single-stranded RNA virus with a positive-stranded genome approximately 28000 nucleotides [6].”

[6]     Chen J, Wang C, Shi H, et al. Molecular epidemiology of porcine epidemic diarrhea virus in China. Archives of virology. 2010;155(9):1471-1476.

  1. Please use italics in the names of viral taxa where needed

ReplyWe made the relevant changes in lines 36, 51-57.

  1. Section 2.5. - please indicate software versions including iTOL and Clustal

ReplyWe have added information about the iTOL software on line 163; the ClustalX software covered in the text is an algorithm used in the MEGA 6.06 software to construct phylogenetic trees, and is not a separate Clustal software, and therefore no information is available about it.

  1. Section 2.5 - please indicate the date or versions of databases used

ReplyWe modified line 159 to add the version number of Genbank, and the result is as follows:“To conduct a phylogenetic analysis, we retrieved the S gene of PEDV strains from GenBank (revision 247.0 ) (Table S1).”

  1. Section 3.4 - Phylogenetic analysis would benefit from comparing the trees constructed using also separate genes. You could talk about recombinations more reasoned

ReplyWe added a related note at line 430-432: " To delve into the predominant genotypes and the spectrum of genotypic diversity among the epidemic strains from 2011 to 2022, we have developed a phylogenetic tree based on the S proteins of PEDV" and modified the reorganization at lines 466-470 as follows:” Zhang et al. identified Poland, Romania, Japan, Thailand, Mexico and Colombia as countries exporting PEDV in Europe, Additionally, Japan plays a significant role as a major import/export centre for PEDV. Moreover, the characteristics of China's PEDV import/export centers indicate the existence of significant transmission links to 11 provinces [44].”

  1. Again, I do not understand which results indicated recombinations.

ReplyIn our study due to the large number of results obtained, there is no way to put them in the article in the form of pictures, we can upload the results in the form of attachments, the reorganization results are shown in Table S9 in the attachment.

  1. Section 3.5 - Perhaps, the table containing a list of alterations would simplify reading.

ReplyWe modified 3.5 in lines 302-304 to read "The analysis of the S protein modelling revealed that the structure of the PEDV S protein underwent changes on an annual basis, exhibiting a discernible pattern of alteration(Figure 7,Table 2)." and added Table 2 in line 329.

Table 2. Statistics on year-to-year changes in PEDV S protein additions

Year of comparison

NTDaa

RBDaa

CV777-HEB(2022)

55-72、85-117、137-140、156-162、188、199-210、211-231

528-536

2021-HEB(2022)

55-64、114-117、132-141、151-164、184、199-201、209-213

563-565

2020-2021

55-59、65-68、111-113、132-135、150-160、184、195-197、206-218

520-529、556-557

2019-2020

157-160

-

2018-2019

157-160

520-553

2017-2018

57-64、68-72、115-117、138-141、156-164、199-201、210-225

524-530

2016-2017

55-60、65-68、111-113、132-135、151-160、195-197、205-219

520-527

2015-2016

56-61、112-114、134-135、207-211

526

2014-2015

56-61、112-114、134-135、207-211

526

2013-2014

52-56、65-68、111-113、126-135、153-159、196-198、206-220

520-529、556-557

2012-2013

38-40、43-51、56-76、84-88、99-102、115-117、133-147、155-161、177-179、184-185、197-202、209-217

522-532、560-567

  1. Figure 7 - I really do not understand why you used homology modeling while AlphaFold is convenient, simple and gives much more precise results.

Reply We sincerely appreciate your advice. However, the data analysis for this article was conducted in 2022, when we were only exposed to Swiss model homology modeling as an approach, and not yet to AlphaFold. But this is really an amazing tool for us to learn and we will use of AlphaFold in the coming research. Thanks again for your nice advice.

  1. Line 412 - please replace “beta” with a greek letter

ReplyWe have modified it in line 409 by changing by it to the Greek letter β.

Round 2

Reviewer 2 Report

Comments and Suggestions for Authors

The quality of the pictures is so bad, that this cannot be published.

Comments on the Quality of English Language

The quality of the pictures is so bad, that this cannot be published.

Author Response

Thank you very much for the reviewing process of our manuscript entitled “Investigation of transmission and evolution of PEDV variants and co-infections in Northern China over past decades” (Manuscript Number: 3088549). We deeply value the reviewer's meticulousness, conscientious feedback, and constructive suggestions. All of these comments are immensely valuable and provide significant guidance for refining and enhancing our paper, as well as steering our research in the right direction. We sincerely appreciate the reviewer's dedicated efforts and earnestly hope that the revisions meet with approval.

We have revised the manuscript accordingly, with yellow highlighting in the revision section of the paper and blue highlighting in the image and table revision section, and detailed point-by-point revisions are listed below:

If your preview of the Word version of the manuscript appears to have a line number that does not match the reply information, please refer to the PDF version of the manuscript.

We have also uploaded a Word version of the reply for you to view, as the content of the reply we have copied into the text box does not display an image.

Reply to reviewer#2

  1. The quality of the pictures is so bad, that this cannot be published.

ReplyWe apologize for the lack of clarity in Figure 3(a) on line 268 of our previous revision. Based on your suggestion, we have adopted a new method to present the amino acid sequences of the PEDV S protein from 2011 to 2022 to ensure clearer and more comprehensible results. The revised result is as follows:

(a)

Figure 3. Sequence analysis of S proteins of the identified PEDV strains from 2011 to 2022. (a)The amino acids of 732 prevalent PEDV S genes from 2011 to 2022 in NCBI Genbank, and 155 S genes identified in this study were used to analyze the amino acid mutation of S protein of PEDV strains: The lines with different colors in the figure represent the corresponding years of the strains, and the rising positions of the lines indicating the amino acid mutation positions. Rising arrows indicate an increase in the probability of co-mutation, while falling arrows indicate a decrease in the probability of co-mutation. The gray underlined letters are the CV777 sequences, the colored letters illustrate the shared sequences of the representative strains in all the years, and the reduced letters depict the mutation patterns.

Reviewer 3 Report

Comments and Suggestions for Authors

The revised version of the manuscript addressed the comments of the previous round. The “Discussion” section was improved and it is more interesting.

Other suggestions were added to improve the manuscript and are shown below:

- Line 17: replace the term “alternatons”.

- Lines 51-68: this paragraph should be presented after paragraphs corresponding to the PEDV explanation (lines 69-114).

- Figure 2: The figure caption for blue and red bars was improved but the explanation contained in the figure was not changed and is confusing.

- Figures 3, 5 and 6; Tables 1 and 2: Detailed Figure caption and table title, adding complete information, are strongly recommended. It is not clear the number of samples that were analyzed and their origin and also the reference sample used without reading the main text. These improvements will help the readers to understand the results presented by the authors.

Additional comment for Figure 5; Tables 1 and 2: the term “Statistics” presented in the title is inadequate because no statistical analyses were performed for data presented in Figure 5 and Tables 1 and 2.

- Figure 7: Figure caption contains repetitive information from the letter (a) to (l) and should be summarized by describing the common information first and then specific information.

Author Response

Thank you very much for the reviewing process of our manuscript entitled “Investigation of transmission and evolution of PEDV variants and co-infections in Northern China over past decades” (Manuscript Number: 3088549). We deeply value the reviewer's meticulousness, conscientious feedback, and constructive suggestions. All of these comments are immensely valuable and provide significant guidance for refining and enhancing our paper, as well as steering our research in the right direction. We sincerely appreciate the reviewer's dedicated efforts and earnestly hope that the revisions meet with approval.

We have revised the manuscript accordingly, with yellow highlighting in the revision section of the paper and blue highlighting in the image and table revision section, and detailed point-by-point revisions are listed below:

If your preview of the Word version of the manuscript appears to have a line number that does not match the reply information, please refer to the PDF version of the manuscript.

We have also uploaded a Word version of the reply for you to view, as the content of the reply we have copied into the text box does not display an image.

Reply to reviewer 3

1.Other suggestions were added to improve the manuscript and are shown below:

- Line 17: replace the term “alternatons”.

ReplyWe apologize for the spelling mistake of the word 'alternaton'. We have revised the original text in line 17, and show it as follows:"

“This epidemiological survey in northern China from 2015 to 2022 examined the prevalence of enteroviruses co-infected with PEDV and the structural mutations in the spike(S)protein resulting from mutations of S gene from 2011 to 2022.”

2.Lines 51-68: this paragraph should be presented after paragraphs corresponding to the PEDV explanation (lines 69-114).

ReplyThank you for the suggestion, and we have placed the paragraph on lines 100-116 to describe the critical role of the PEDV S protein in viral binding and host invasion, and the contribution of its mutations to genetic diversity and changes in virulence first, and summary of the current common viral causes of swine diarrhea and co-infections in China

3.Figure 2: The figure caption for blue and red bars was improved but the explanation contained in the figure was not changed and is confusing.

ReplyWe have rewritten the graphical description on line 211 of Figure 2 as follows: the blue bar is " Prevalence of mixed infections with viruses shown on the horizontal axis in samples that tested positive for PEDV," and the red bar is "Prevalence of mixed infections with PEDV in samples that tested positive for the viruses shown on the horizontal axis."

Figure 2. Co-infection of PEDV with selected enteroviruses in diarrhea samples from 2015 to 2022. The blue bar represents the prevalence of PEDV in samples that tested positive for PEDV and other viruses. The prevalence of mixed infection with PEDV for each of the other 11 diarrhea viruses is indicated by the red bar.

4.Figures 3, 5 and 6; Tables 1 and 2: Detailed Figure caption and table title, adding complete information, are strongly recommended. It is not clear the number of samples that were analyzed and their origin and also the reference sample used without reading the main text. These improvements will help the readers to understand the results presented by the authors.

Additional comment for Figure 5; Tables 1 and 2: the term “Statistics” presented in the title is inadequate because no statistical analyses were performed for data presented in Figure 5 and Tables 1 and 2.

ReplyWe appreciate your advice. We have revised the figure captions and table captions of Figure 3(in line 272-284), Figure 5(in line 315) and Figure 6(in line 333); Tables 1(in line 285) and Table 2 (in line 356), and added the relevant information, as follows:

Figure 3. Sequence analysis of S proteins of the identified PEDV strains from 2011 to 2022. (a)The amino acids of 732 prevalent PEDV S genes from 2011 to 2022 in NCBI Genbank, and 155 S genes identified in this study were used to analyze the amino acid mutation of S protein of PEDV strains: The lines with different colors in the figure represent the corresponding years of the strains, and the rising positions of the lines indicating the amino acid mutation positions. Rising arrows indicate an increase in the probability of co-mutation, while falling arrows indicate a decrease in the probability of co-mutation. The gray underlined letters are the CV777 sequences, the colored letters illustrate the shared sequences of the representative strains in all the years, and the reduced letters depict the mutation patterns. (b) Annual variation statistics of amino acid mutations in S proteins of all selected PEDV strains (n=732+155) prevalent worldwide from 2012 to 2022. The vertical coordinate indicates the number of mutations, the horizontal coordinate is the year of PEDV prevalence, and the dots shown in the violin plot indicate the number of mutations in the S protein of PEDV strains prevalent in each year.

Figure 5. Recombination positions of global PEDV prevalent strains (n=732+155) from 2011 to 2022.

Figure 6. S gene-based phylogenetic analysis of northeast China endemic PEDV strains (n=155). The colors indicate the year in which the strains were identified, and the lines in the outer circle of the evolutionary tree represent the subtypes of the strains.

Table 1. New mutations in the S protein of representative PEDV strains between 2011 and 2022.

Table 2. The year-on-year range of structural residues of protein S of the representative PEDV strains.

  1. Figure 7: Figure caption contains repetitive information from the letter (a) to (l) and should be summarized by describing the common information first and then specific information.

Reply We have rewritten the labeling of Figure 7 in line 350 as you suggested, and the revised result is as follows:

Figure 7. Year-on-year comparison of the S protein modeling differences between 2011 and 2022 prevalent PEDV strains.

(a) 2011 and 2012_PEDV strains; (b) 2012 and 2013_PEDV strains; (c) 2013 and 2014_PEDV strains; (d) 2014 and 2015_PEDV strains; (e) 2015 and 2016_PEDV strains; (f) 2016 and 2017_PEDV strains; (g) 2017 and 2018_PEDV strains; (h) 2018 and 2019_PEDV strains; (i) 2019 and 2020_PEDV strains; (j) 2020 and 2021_PEDV strains; (l) CV777 and 2022_PEDV strains.
